# External Validation of an Artificial Intelligence Triaging System for Chest X-Rays: A Retrospective Independent Clinical Study

**DOI:** 10.3390/diagnostics15222899

**Published:** 2025-11-15

**Authors:** André Coutinho Castilla, Iago de Paiva D’Amorim, Maria Fernanda Barbosa Wanderley, Mateus Aragão Esmeraldo, André Ricca Yoshida, Anthony Moreno Eigier, Márcio Valente Yamada Sawamura

**Affiliations:** 1NeuralMed A2 Tecnologia Ltda., Rua Pe João Manoel 1212, Cj 41, São Paulo 01411-010, SP, Brazil; nandaw@gmail.com (M.F.B.W.); andre.yoshida@neuralmed.com (A.R.Y.); a@neuralmed.com (A.M.E.); 2Department of Radiology and Oncology, Hospital das Clínicas, Faculdade de Medicina, Universidade de São Paulo (HCFMUSP), Av. Dr. Enéas de Carvalho Aguiar 255, São Paulo 05403-000, SP, Brazil; iago.damorim@hc.fm.usp.br (I.d.P.D.); marcio.sawamura@hc.fm.usp.br (M.V.Y.S.); 3Department of Radiology, Stanford University School of Medicine, Palo Alto, CA 94305, USA; mateusesmeraldo3@gmail.com

**Keywords:** artificial intelligence, chest radiography, triage, external validation, deep learning, radiology, diagnostic accuracy, medical imaging

## Abstract

**Background**: Chest radiography (CXR) is the most frequently performed radiological exam worldwide, but reporting backlogs, caused by a shortage of radiologists, remain a critical challenge in emergency care. Artificial intelligence (AI) triage systems can help alleviate this challenge by differentiating normal from abnormal studies and prioritizing urgent cases for review. This study aimed to externally validate TRIA, a commercial AI-powered CXR triage algorithm (NeuralMed, São Paulo, Brazil). **Methods**: TRIA employs a two-stage deep learning approach, comprising an image segmentation module that isolates the thoracic region, followed by a classification model trained to recognize common cardiopulmonary pathologies. We trained the system on 275,399 CXRs from multiple public and private datasets. We performed external validation retrospectively on 1045 CXRs (568 normal and 477 abnormal) from a teaching university hospital that was not used for training. We established ground truth using a large language model (LLM) to extract findings from original radiologist reports. An independent radiologist review of a 300-report subset confirmed the reliability of this method, achieving an accuracy of 0.98 (95% CI 0.978–0.988). We compared four ensemble decision strategies for abnormality detection. Performance metrics included sensitivity, specificity, accuracy, and area under the receiver operating characteristic curve (AUROC) with 95% CI. **Results**: The general abnormality classifier achieved strong performance (AUROC 0.911). Individual pathology models for cardiomegaly, pneumothorax, and effusion showed excellent results (AUROC of 0.968, 0.955, and 0.935, respectively). The weighted ensemble demonstrated the best balance, with an accuracy of 0.854 (95% CI, 0.831–0.874), a sensitivity of 0.845 (0.810–0.875), a specificity of 0.861 (0.830–0.887), and an AUROC of 0.927 (0.911–0.940). Sensitivity-prioritized methods achieving sensitivity >0.92 produced lower specificity (<0.69). False negatives were mainly subtle or equivocal cases, although many were still flagged as abnormal by the general classifier. **Conclusions**: TRIA achieved robust and balanced accuracy in distinguishing normal from abnormal CXRs. Integrating this system into clinical workflows has the potential to reduce reporting delays, prioritize urgent cases, and improve patient safety. These findings support its clinical utility and warrant prospective multicenter validation.

## 1. Introduction

Chest radiography (CXR) is the most common radiological examination performed globally and serves as a critical first-line diagnostic tool [1]. However, the growing global demand for imaging, combined with an ongoing shortage of radiologists, often results in substantial reporting backlogs. These delays are particularly critical in emergency department settings, where rapid interpretation is essential for patient management. In many settings, general practitioners and physicians often perform the initial interpretation of CXRs instead of radiologists, which can lead to diagnostic errors or delays due to a potential lack of specialized expertise.

Artificial intelligence (AI) has emerged as a promising approach to addressing these challenges in healthcare. The advent of deep learning, specifically in the form of convolutional neural networks, has enabled the development of diagnostic models that achieve performance levels comparable to those of expert clinicians in various medical imaging tasks [2]. Among the possible impactful applications of AI in radiology is triage, the automated prioritization of medical cases to ensure that critical or abnormal studies receive earlier review by radiologists [3]. A system that can reliably distinguish between normal and abnormal CXRs may enhance physicians’ ability to detect abnormalities [4], reduce diagnostic delays, and optimize clinical workflows by directing urgent examinations for immediate specialist evaluation.

This growth has driven substantial investment and regulatory interest in AI-based radiological tools. Radiology now represents the medical specialty with the highest number of approved AI software devices worldwide [5]. As a consequence, rigorous external validation and post-implementation assessment have become imperative to ensure these algorithms perform safely and effectively in diverse real-world clinical environments.

This study aims to conduct an external validation of TRIA, a commercial AI-powered CXR triage system developed by NeuralMed (NeuralMed, São Paulo, Brazil) and registered with the Brazilian Health Regulatory Agency (ANVISA) as a Class II medical device (Register No. 8261971001) [6]. We retrospectively evaluated its performance at the Hospital das Clínicas, Faculty of Medicine, University of São Paulo (HCFMUSP), to determine its diagnostic accuracy in classifying CXRs across seven distinct radiologic findings and in differentiating normal from abnormal studies. We compared the algorithm’s predictions with radiologist reports, providing an objective assessment of its clinical reliability. Evaluating algorithmic performance against verified ground truth diverse from its training dataset is a critical step in demonstrating clinical utility and regulatory compliance.

In this article, we describe the development of the TRIA system, its dataset composition, and the evaluation methodology. This work supports ANVISA’s requirements for manufacturers to maintain continuous clinical evaluation and post-market surveillance of medical AI devices [7]. NeuralMed funded the study, and the institutional Ethics Committee approved it. The specific aims were to evaluate the overall diagnostic accuracy of TRIA’s general abnormality classifier in a dataset independent of that used in training; assess the performance of individual classifiers for key pathologies (e.g., cardiomegaly, pneumothorax, and pleural effusion); compare four ensemble strategies for abnormality detection to identify the optimal balance between sensitivity and specificity for triage use; conduct an error analysis of false negatives to characterize the system’s limitations and potential areas for improvement; and compare its performance with similar available tools.

## 2. Methods

### 2.1. AI Algorithm (TRIA)

The algorithm was based on a two-stage deep learning approach, which consisted of a lung field segmentation pipeline followed by a pathology classification model. A key design choice in the model’s methodology was to prioritize intrathoracic findings related to the cardiopulmonary system, while deprioritizing musculoskeletal and spinal conditions.

#### 2.1.1. Training Dataset

We developed the models using a large-scale, multi-source dataset of 275,399 anonymized, unique chest radiographs. We curated the dataset from public and private sources and featured a multi-label annotation schema (Table 1). Part of the dataset originated from private clinical sources collected under research and commercial agreements in compliance with Brazilian General Data Protection Law [8].

The public datasets included: the Shenzhen Chest X-ray Set [9]; the Indiana University Chest X-ray Collection [10]; the JSRT database from the Japanese Society of Radiological Technology [11]; the NIH ChestX-ray14 dataset [12]; and the PadChest dataset [13]. We utilized all publicly available datasets in accordance with their respective open-access agreements.

Before training, radiologists reviewed and verified the images, applying annotations using a proprietary software application (Figure 1). The dataset exhibited a significant class imbalance, reflecting the real-world prevalence of the pathology. For training binary classifiers, studies with definitive labels (1.0 for present, 0.0 for absent) were used. In contrast, studies with missing labels for a specific finding were excluded from the loss calculation for that task. Table 2 shows the distribution for a selection of key labels. The list of pathologies was selected based on the availability of labeled data, balanced with clinical importance for the ED. The ‘normal’ and ‘abnormal’ pools included cases with pathologies beyond the specifically targeted list; these were categorized under the general “yesfinding” class.

#### 2.1.2. Algorithm Architecture

The proposed system consisted of two main stages: (1) segmentation of the thoracic region and (2) classification of pathological findings. This modular design improved the model’s interpretability, efficiency, and diagnostic precision by separating spatial localization from disease recognition.

We employed a U-Net architecture with a MobileNetV2 backbone to segment the thoracic region and generate a unified “tongue-shaped” mask encompassing both lungs as well as the mediastinal and hilar regions. We chose U-Net for its proven effectiveness in medical image segmentation tasks, where high spatial precision is required. Its encoder–decoder “U” structure enabled the capture of both contextual and spatial information, facilitating the delineation of complex anatomical structures with high resolution. In this study, segmentation was used not only to isolate the relevant pulmonary region but also to constrain the subsequent classifier’s focus to clinically meaningful areas, thereby improving accuracy and reducing noise from irrelevant image regions.

We resized input images to 512 × 512 pixels and performed data augmentation using geometric and photometric transformations (padding, Gaussian noise, CLAHE, brightness/gamma adjustment, blur, motion blur, and hue/saturation shifts). The model was trained for 40 epochs with a batch size of 8 using the Adam optimizer (initial learning rate = 5 × 10^−5^) and a composite loss function combining Dice loss and binary focal loss to balance region overlap and class imbalance. Performance was monitored using Intersection over Union and F-score metrics, and the best model was selected by minimizing the validation loss with adaptive learning rate scheduling. Figure 2 illustrates the resulting mask, which ensures inclusion of central thoracic structures.

The second stage employed an InceptionResNetV2 convolutional neural network, pre-trained on ImageNet, adapted for binary and multi-class classification of selected thoracic abnormalities. The model’s top layers were replaced with custom fully connected layers, optimized for CXR analysis. To balance computational efficiency with diagnostic performance, input images were resized to 75 × 75 pixels. Data augmentation included random brightness (range 0.3–0.5), rotation (≤35°), zoom (≤12.5%), and rescaling (1/255). Training utilized the Adam optimizer (learning rate = 1 × 10^−4^) with binary cross-entropy loss and class weighting to address data imbalance. There were also callbacks for stopping training when no improvement occurred for five epochs, saving the best model during training, and reducing the learning rate when performance plateaued for two epochs. The model’s performance was evaluated using accuracy, precision, recall, and area under the receiver operating characteristic curve (AUROC), incorporating Test-Time Augmentation for robust estimation. In this phase, we generated the classification thresholds. All experiments were tracked using MLflow to ensure reproducibility.

To enhance model transparency and validate spatial reasoning, we applied Gradient-weighted Class Activation Mapping (Grad-CAM) to visualize the regions most influential in classification decisions. Grad-CAM served a dual purpose in this context: first, it verified that the model’s predictions are based on clinically relevant regions within the segmented thoracic area; second, it contributed to interpretability by generating intuitive heatmaps that highlight the anatomical structures driving the model’s output. This step was crucial for improving classification accuracy and verifying that it relied on the correct visual cues, aligning with radiological expectations. For optimization, they were only generated when the classification probability score was above the defined threshold.

Representative Grad-CAM overlays are shown in Figure 3, Figure 4, Figure 5 and Figure 6, illustrating several different types of pathologies. These visualizations demonstrate where the algorithm focuses attention on the appropriate pathological regions, thereby reinforcing the validity and interpretability of the proposed pipeline.

All models were developed in Python (Version 3.6.9) using TensorFlow and Keras, with image manipulation performed using OpenCV. Training and validation were executed on GPU hardware to ensure stable convergence and reproducibility.

### 2.2. Validation Study Design

For validation, we conducted a retrospective, cross-sectional study on a dataset of 1045 anonymized chest radiographs from HCFMUSP, acquired over the past five years. Eligibility was restricted to adult CXR studies that included both frontal and lateral images and had an accompanying free-text report authored by a senior radiologist. To ensure a comprehensive evaluation of the algorithm’s performance on both normal and abnormal cases, we employed a purposive sampling strategy. We constructed the final cohort by selecting 568 examinations with reports describing a normal CXR and 477 examinations with abnormal findings. The abnormal cases were identified by selecting reports that lacked a normal description and contained one or more of the following keywords: “pneumonia”, “opacity”, “cardiomegaly”, “pneumothorax”, “effusion”, “nodule”, or “mass”. The validation cohort consisted of adult patients with a mean age of 47.9 years (median, 52 years; standard deviation, 18.4; range, 19–82 years). There was a balanced sex distribution, with 495 female patients (47%) and 550 male patients (53%).

To generate structured, binary labels for the quantitative evaluation, we processed the narrative reports using a sophisticated Large Language Model (LLM) pipeline designed to transform unstructured text into a predefined, machine-readable format [14]. The contribution of the LLM extends significantly beyond simple label extraction; it functions as a data structuring engine. This was achieved by providing the model with a detailed, multi-component prompt that meticulously guided its output. The prompt included: (1) explicit rules for generating a valid JSON object, such as the mandatory use of double quotes and binary indicators for the presence or absence of a condition; (2) precise classes with definitions for each of the target pathologies; (3) a clear output template specifying the desired JSON schema; and (4) illustrative examples of reports paired with their correct JSON counterparts to provide concrete guidance.

This structured prompting strategy compelled the LLM to interpret the radiological findings within the context of the provided definitions and rules, thereby producing a consistent and analysis-ready dataset. The resulting JSON from the LLM contained the structured labels used to evaluate the normal and abnormal separation, the accuracy of each pathology classifier, and the criticality ensemble. We used only the extracted labels to assess the algorithm; the Grad-CAM images are presented here for illustration purposes only. All radiographs were processed by the TRIA system, which served the institution through its Picture Archiving and Communication System (PACS), utilizing files in Digital Imaging and Communications in Medicine (DICOM) format.

### 2.3. Analysis

We evaluated a two-stage computational pipeline designed to optimize individual pathology classifiers first and then assess the performance of various ensemble strategies.

In the initial stage, we determined optimal classification thresholds for each pathology. We achieved this by generating a receiver operating characteristic (ROC) curve for each classifier against its corresponding ground truth label. We then identified the optimal probability threshold using Youden’s J statistic. This method maximizes the sum of sensitivity and specificity, thereby finding an optimal balance between true positives and true negatives [15].

In the second stage, we implemented and evaluated four distinct ensemble strategies to aggregate the outputs from the individual classifiers into a single, study-level determination of “normal” or “abnormal.” The specific logic for each strategy is detailed below:Possibility-based (Rule-based Commercial Algorithm): This method emulates the logic of the commercial product. It classifies an examination as abnormal if any pathology is assigned a qualitative likelihood of high or certain. This approach relied on pre-defined categorical bins derived from probability scores for its final decision.Threshold-based: This strategy classified an examination as abnormal if the predicted probability score for any individual pathology exceeded its previously determined optimal threshold from the first stage of analysis. This strategy tests a straightforward aggregation rule where any single positive finding renders the entire study abnormal.Weighted Sum-based: This strategy computed a weighted sum of the probability scores from all classifiers for a given study. To emphasize the importance of the global abnormality detector, we assigned the score from the general yesfinding classifier a weight of 2.0. In contrast, the scores for all other specific pathologies were assigned a weight of 1.0. We classified an examination as abnormal if this aggregated weighted sum surpassed a unique, optimized threshold determined via ROC analysis on the sum scores themselves. This explicit up-weighting of the yes-finding output was the basis for its “weighted” designation.Hierarchical-based: This model used a multi-step, rule-based logic that prioritized specific findings. First, the yesfinding score was evaluated: if it is significantly below its optimal threshold (e.g., <0.8× threshold), the model classified the exam as normal. Conversely, if the score is significantly greater than high (e.g., >1.2× threshold), it flags the case as abnormal. For scores falling within this intermediate range, the algorithm proceeded to check for the presence of findings. If any of these are predicted with a score above their respective optimal thresholds, the model classifies the examination as abnormal. Otherwise, the final decision defaults to yes-finding, based on the yes-finding score, which is compared against its standard threshold.

To determine whether the observed differences between the performing ensemble strategies were statistically significant, we employed two complementary tests: the DeLong test for comparing areas under the AUROC curve and McNemar’s test for paired categorical outcomes (accuracy, sensitivity, and specificity). A *p*-value < 0.05 was considered statistically significant. We performed all analyses using paired predictions from the same validation dataset to ensure direct comparability across models.

## 3. Results

### 3.1. Individual Classifier Performance

The proposed TRIA system demonstrated strong discriminative performance across multiple thoracic pathologies. The general “yesfinding” classifier achieved an AUROC of 0.911, indicating robust overall abnormality detection. Among the individual models, cardiomegaly, pneumothorax, and pleural effusion achieved excellent diagnostic accuracy, with AUROC values of 0.968, 0.955, and 0.935, respectively. The consolidation category in prior studies showed high sensitivity (0.829) but comparatively lower specificity (0.656), reflecting the smallest training dataset. Detailed results for all classifiers are provided in Table 3.

### 3.2. Ensemble Model Performance for Abnormality Detection

Among the four ensemble approaches, the weighted sum-based model demonstrated the best balance between sensitivity and specificity, achieving an accuracy of 0.854 (95% CI: 0.831–0.874), sensitivity of 0.845 (95% CI: 0.810–0.875), specificity of 0.861 (95% CI: 0.830–0.887), and AUROC of 0.927 (95% CI: 0.911–0.940). De Long’s test confirmed that its AUROC was significantly higher than that of the threshold-based model (p=0.0307).

Paired McNemar’s tests further revealed that the weighted sum-based ensemble significantly outperformed the threshold-, possibility-, and hierarchical-based methods in terms of overall accuracy (p<0.01 for all comparisons) and specificity (p<0.001), while maintaining comparable sensitivity (p>0.05 when compared with the hierarchical model) (Table 4). These findings indicate that the weighted aggregation approach achieves superior discrimination and generalization, with statistically validated improvements in diagnostic performance. Comprehensive results are presented in Table 5, and corresponding receiver operating characteristic curves and confusion matrices are illustrated in Figure 7 and Figure 8.

### 3.3. Error Analysis

An analysis of the algorithm’s false negatives revealed a nuanced performance profile, where a specific miss did not always equate to a complete system failure. Across the board, a substantial number of cases with a false negative for a particular pathology were nevertheless correctly identified as abnormal by the general “yes-finding” classifier or had other co-occurring true positive pathologies. For example, out of 38 false negatives for opacity, the algorithm correctly identified the exam as abnormal in 25 cases. In many instances of a missed finding, the algorithm successfully detected other pathologies within the same study. Misclassification was another notable source of error, where an existing pathology was detected but incorrectly labeled, such as opacities being predicted as masses or nodules (Figure 4). Radiologist review of these false-negative cases frequently noted findings that were “subtle,” “small,” “doubtful,” “seen on lateral image only,” or “non-specific.” In other cases, the algorithm correctly identified a more critical finding, like a large pleural effusion, while missing a secondary, less significant pathology (Figure 3). Figure 9, Figure 10 and Figure 11 illustrate false negative cases for opacity, pneumothorax, and nodule, respectively. This highlights the inherent limitations and inevitable failures of this kind of system.

### 3.4. Clinical Workflow

As previously stated, clinicians implement TRIA in a clinical scenario. Upon image acquisition, CXRs are automatically forwarded from the institution’s PACS to the TRIA server. The algorithm processes the study and sends its findings—including the classification derived from the possibility ensemble and visual explainability aids, such as Grad-CAM heatmaps—directly back to the PACS. These results are then displayed to the radiologist within their standard viewer, alongside the original image (Figure 12). This seamless integration embeds the entire analysis within the patient’s study, enabling immediate worklist prioritization and concurrent review by the attending physician.

## 4. Discussion

The intended clinical application for the TRIA system is as a triage tool to prioritize patients, a typical and valuable use case for radiology AI [3,16]. This study provides a necessary external validation of the system for this purpose. The evaluation methodology we used is a standard approach in AI validation, as highlighted in systematic reviews and other external evaluations [16,17,18].

A key methodological strength of the underlying AI model is the use of a tongue-shaped segmentation mask. By including the mediastinum and hilar regions in the ROI, the model analyzes these diagnostically critical areas, which are often excluded by traditional lung-only segmentation. This approach creates a more clinically relevant area for analysis, which may improve the detection of central pathologies, especially those related to the cardiopulmonary system.

However, a significant trade-off in the model’s design is the use of a very low input resolution (75 × 75 pixels) for classification. While this approach allows for computational efficiency, it inherently risks the loss of fine-grained details necessary for detecting subtle pathologies, such as small nodules or early interstitial disease. This was reflected in the error analysis, where false negatives were often subtle cases. The high false-positive rate for conditions like opacity and consolidation suggests the model may be oversensitive to non-specific patterns, highlighting a discrepancy between pure pattern recognition and a radiologist’s clinical interpretation, which incorporates a higher threshold for significance.

The set of pathologies chosen to build the tool reflects the importance of critical findings for emergency physicians who are not assisted by radiology specialists. The limited number of specific pathologies favors computational efficiency, while the general normal/abnormal classifier (“yesfinding”) provides a safeguard for unlisted pathologies, as depicted by the ensemble results.

In real-world clinical settings, the prevalence of abnormal CXR is substantially lower than that in research datasets, often leading to significant class imbalance. This imbalance tends to skew diagnostic performance toward more common findings such as opacities and cardiomegaly, while less frequent pathologies—such as pleural effusion, pulmonary nodules, and pneumothorax—are underrepresented. For instance, unpublished internal data from various initial deployments of the algorithm at a new client site, typically a survey of 300 consecutive examinations, indicate an abnormal-to-normal ratio of approximately 1:5, with several subsets containing no cases of the three less prevalent conditions mentioned earlier. This observation justified the use of a curated dataset for validation, ensuring adequate representation of all target pathologies. Furthermore, it is essential to note that lower disease prevalence intrinsically reduces the positive predictive value and increases the negative predictive value, even when sensitivity and specificity remain constant [19].

From a methodological standpoint, a consecutive prospective dataset could provide higher distributional validity; however, the low frequency of abnormal findings in such samples can limit the precision and statistical power for estimating algorithm accuracy. Therefore, we opted for a balanced and enriched dataset to ensure adequate representation of clinically relevant abnormalities. This lower proportion of normal cases does not compromise the validity of our evaluation, given that the validation dataset comprising 1045 curated CXR provides a statistically robust sample size. This cohort yields 95% confidence intervals with margins of error of approximately ±3–4% for key performance metrics, surpassing typical requirements for medical imaging validation. Nevertheless, we still acknowledge that this enrichment does not replicate true disease prevalence and differs from studies employing large, consecutive, real-world datasets, which provide a more realistic benchmark of clinical performance [20,21].

We established ground truth through LLM–assisted extraction of findings from reports authored by senior radiologists, a pragmatic strategy for large-scale datasets. A radiologist independently reviewed a subset of 300 reports, yielding an accuracy of 0.9838 (95% CI: 0.9783–0.9883), thereby supporting the reliability of the adopted approach. Nevertheless, more rigorous standards for ground truth determination exist, such as consensus interpretation by a panel of expert radiologists or correlation with reference imaging modalities, including computed tomography [20,22,23,24].

A comparative analysis reveals a landscape of varied performance among CXR AI tools. Regarding the individual pathologies classification, we compared TRIA’s performance with CheXNet [25], a well-established deep learning model trained on the NIH ChestX-ray14 dataset and often used as a benchmark for CXR classification. As shown in Table 6, TRIA achieved higher AUROC values for cardiomegaly and pleural effusion compared to CheXNet, while the consolidation and opacity classifiers demonstrated comparable performance.

In this study, the possibility-based ensemble (used in the commercial product) obtained high sensitivity. In contrast, the weighted sum ensemble demonstrated robust and balanced performance for abnormality stratification, achieving an AUROC of 0.927 with a sensitivity of 84.5% and a specificity of 86.1%. This balanced profile contrasts with other AI triage tools, such as the one evaluated by Blake et al., which was optimized for very high sensitivity (99.6%) at the expense of lower specificity (67.4%) [16]. Another algorithm evaluated in a primary care setting showed excellent specificity (92.0%) but significantly lower sensitivity (47.0%) [26]. These differences represent fundamental design choices: one prioritizes minimizing missed findings, accepting more false positives, while the other seeks to balance sensitivity and specificity to reduce workload more efficiently. The overall AUROC for TRIA is comparable to that of high-performing algorithms evaluated in other studies, which often achieve AUROCs in the 0.92 to 0.94 range [27,28].

These results demonstrate that TRIA performs on par with or above the level of reference models in the literature, particularly for acute pathologies most relevant to emergency triage workflows. Importantly, TRIA was externally validated using an independent real-world hospital dataset, thereby reinforcing its generalizability of performance across diverse imaging conditions.

The error analysis of TRIA, which noted that false negatives were often subtle or challenging cases, is a consistent finding across the field. For instance, Plesner et al. similarly found that the performance of four different commercial tools declined for smaller-sized findings and on complex radiographs with multiple pathologies [20]. Our study identifies the model’s low input resolution as a potential factor contributing to the missing of fine-grained details. Despite this, while the model may misclassify or miss a specific finding, its ability to flag an exam as generally abnormal remains high, ensuring that complex cases are still prioritized for radiologist review. The variability highlighted in Table 7 also shows that algorithm performance is highly dependent on the clinical task and validation setting, suggesting that a single set of metrics may not fully capture an algorithm’s utility.

We acknowledge that the primary potential source of conflict in this study is sponsor bias. To minimize this risk, the study was initiated at the request of the participating institution to independently assess the performance of the TRIA algorithm after its clinical deployment. Notably, the validation dataset originated from an external institution, entirely independent from the one involved in TRIA’s development. We derived the ground truth from diagnostic reports authored by senior radiologists at this external center, without any involvement from company personnel. Researchers from HCFMUSP exclusively performed case selection and data extraction, ensuring an impartial and methodologically sound process. We predefined the study protocol and analysis plan before data analysis, thereby reinforcing methodological transparency. All patient data were fully anonymized at the source in accordance with institutional policies and data protection regulations. The institutional review board (IRB) approved the study and waived the requirement for informed consent due to its retrospective, observational design. We extracted data directly from the hospital PACS after the AI system had already been integrated into routine clinical workflows, ensuring real-world representativeness of the validation process.

We also recognize that this validation was conducted at a single tertiary academic hospital, which may limit the generalizability of our findings to other clinical contexts, such as primary care or community hospitals, where patient demographics, equipment, and workflow conditions may differ. Future research should include prospective, multicenter evaluations across diverse healthcare settings to confirm the robustness and clinical impact of TRIA under varying operational conditions. Additionally, this study’s retrospective design may overestimate algorithmic performance compared to real-time prospective use, as reported in prior work on diagnostic AI translation to practice [17]. Finally, the validation dataset was enriched with clinically significant pathologies, which does not reflect the true prevalence of abnormalities in unselected populations and may influence the apparent predictive performance.

Further refinement of the classification model, particularly by adding more pathologies and exploring higher input resolutions, could enhance its ability to detect more subtle findings and improve its clinical utility.

## 5. Conclusions

This independent retrospective validation demonstrates that the TRIA AI algorithm achieves robust and accurate performance in discriminating between normal and abnormal CXR. The actual commercial version has excellent sensitivity and provides confidence for clinical implementation as a reliable tool for triaging CXRs. The strong and balanced performance of the weighted sum ensemble strategy offers a viable alternative when greater specificity is required. The tool could help manage reporting backlogs, prioritize urgent cases for radiologist review, and potentially support earlier patient intervention. This study provides a foundation for future prospective and comparative research to evaluate the algorithm’s clinical utility further.

## Figures and Tables

**Figure 1 diagnostics-15-02899-f001:**
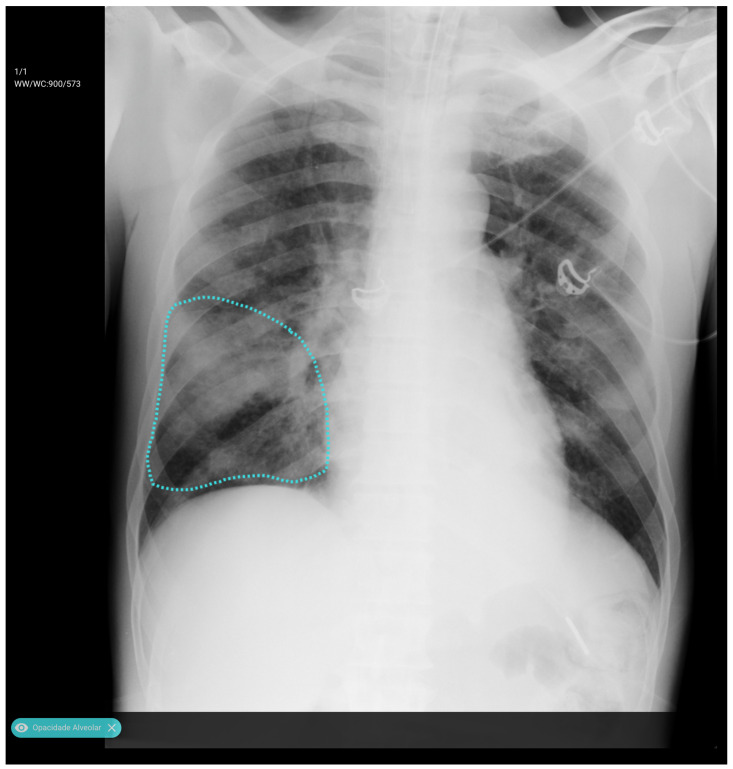
All labeled images were proofed by a radiologist who manually drew a region of interest around the pathology and assigned a finding, as shown here for an alveolar opacity circled by the dashed line.

**Figure 2 diagnostics-15-02899-f002:**
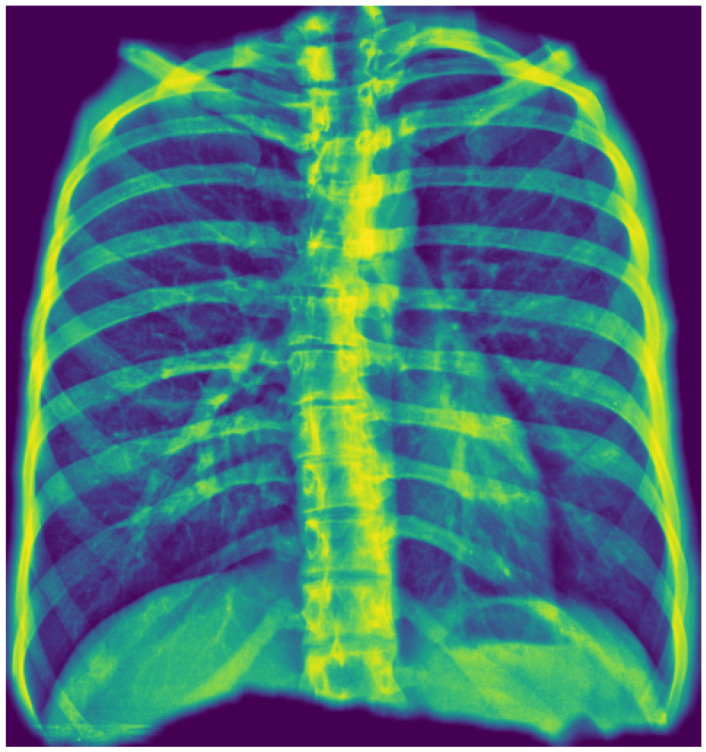
Example of first-stage segmentation: the generated “tongue-shaped” mask covering both lungs and mediastinal structures.

**Figure 3 diagnostics-15-02899-f003:**
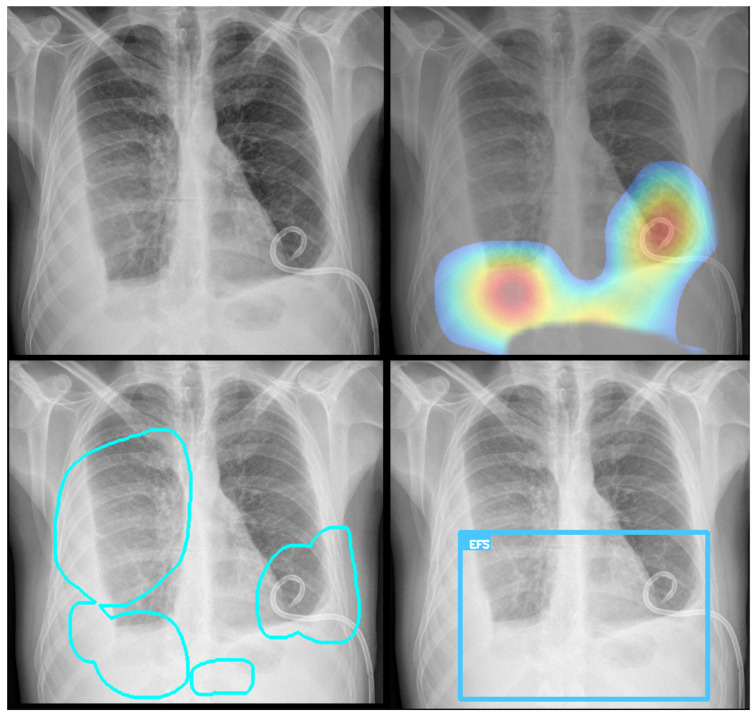
Example of Grad-CAM visualizations localizing detected findings. The figure shows the original image (**top left**) alongside three types of explanatory outputs generated by the algorithm: a heatmap (**top right**), a contour outline (**bottom left**), and a bounding box (**bottom right**). All visualizations are derived from the same process, showcasing different output options.

**Figure 4 diagnostics-15-02899-f004:**
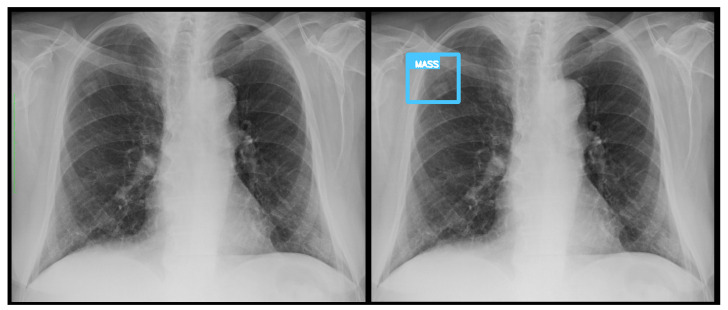
A true positive example of a nodule/mass (**left**) and its bounding box generated image (**right**). In this case, the report indicated opacity; however, the algorithm classified as a mass.

**Figure 5 diagnostics-15-02899-f005:**
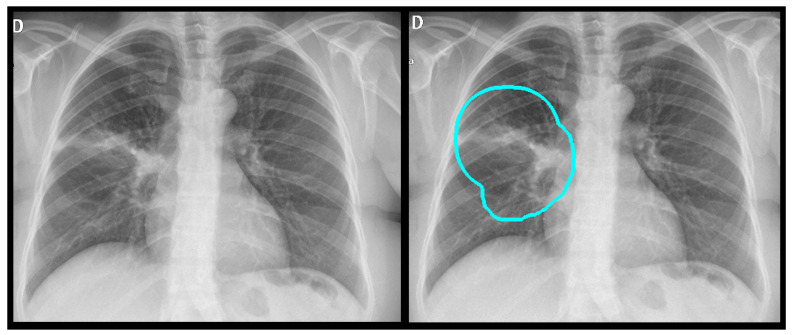
A true positive example of opacity and consolidation (**left**) and its contour line generated image (**right**).

**Figure 6 diagnostics-15-02899-f006:**
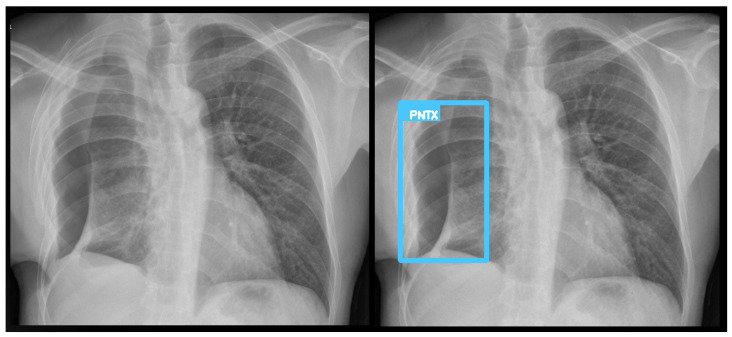
A true positive example of pneumothorax (**left**) and its bounding box generated image (**right**). The algorithm successfully detected pneumothorax without a pleural tube, indicating it is not overfit to post-drainage cases.

**Figure 7 diagnostics-15-02899-f007:**
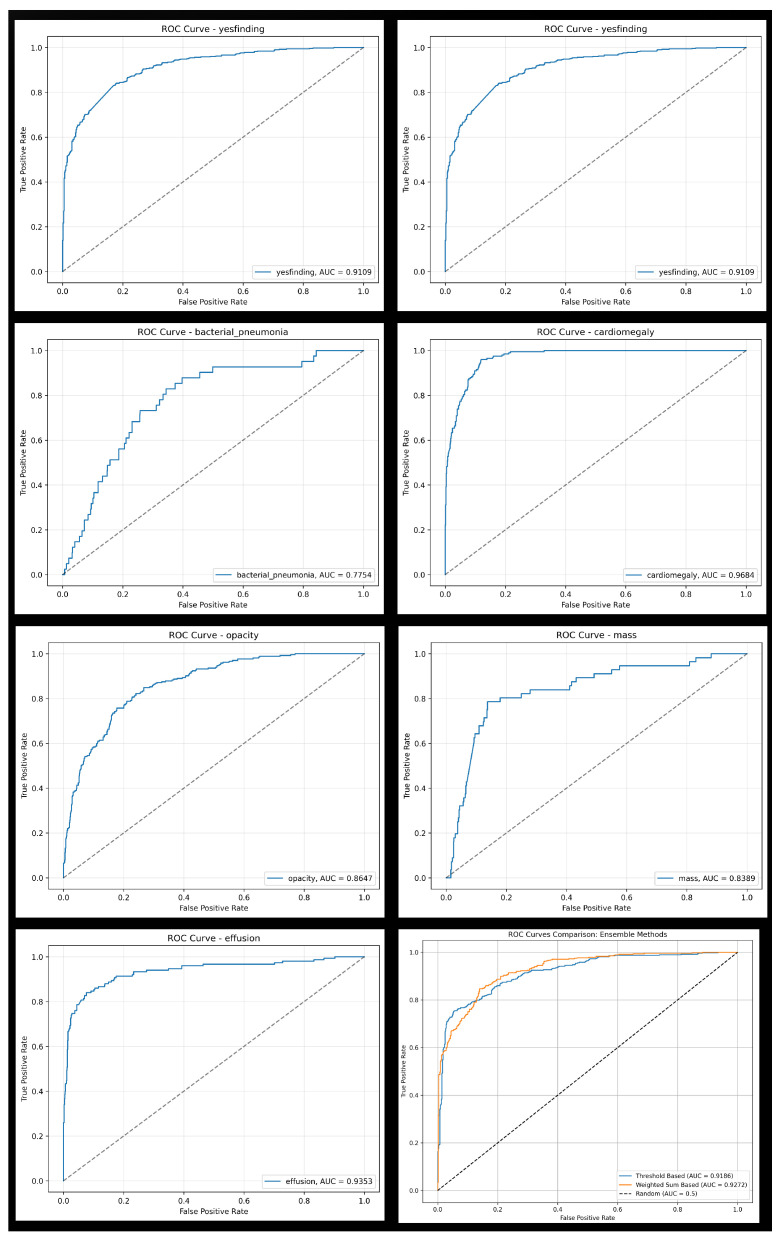
Comparison of receiver operating characteristic curves for the seven selected pathologies. The bottom right panel shows the curves for Threshold-Based and Weighted Sum-Based ensemble methods.

**Figure 8 diagnostics-15-02899-f008:**
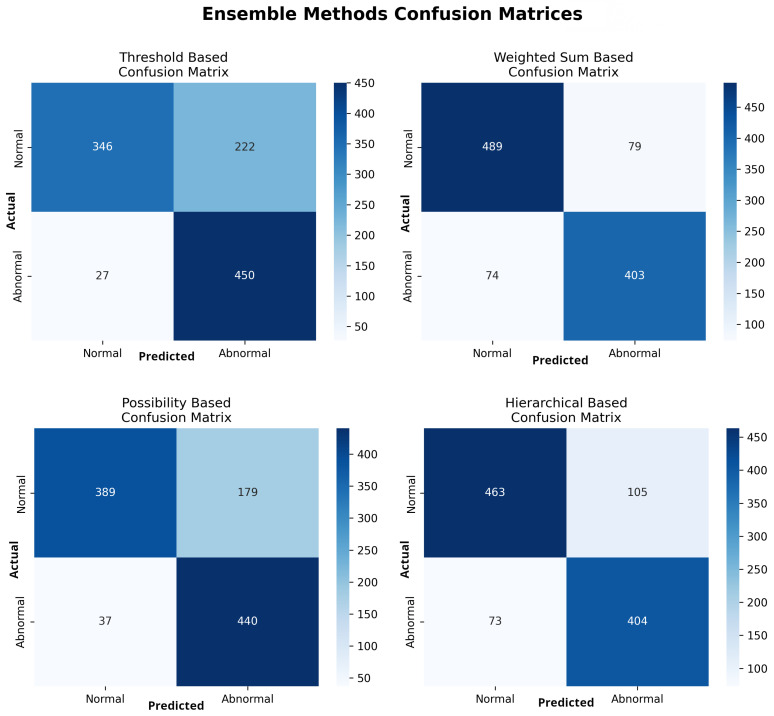
Confusion matrices for the four ensemble methods evaluated for abnormality detection.

**Figure 9 diagnostics-15-02899-f009:**
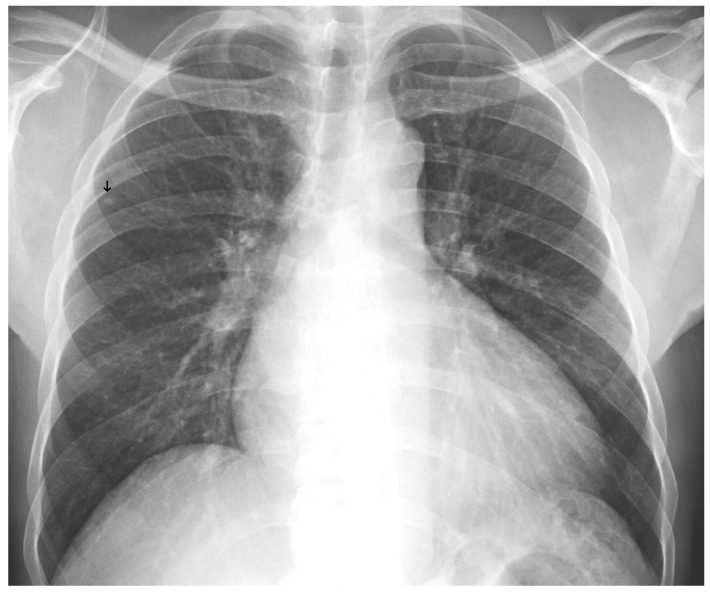
A false negative example of a small, missed nodule (black arrow).

**Figure 10 diagnostics-15-02899-f010:**
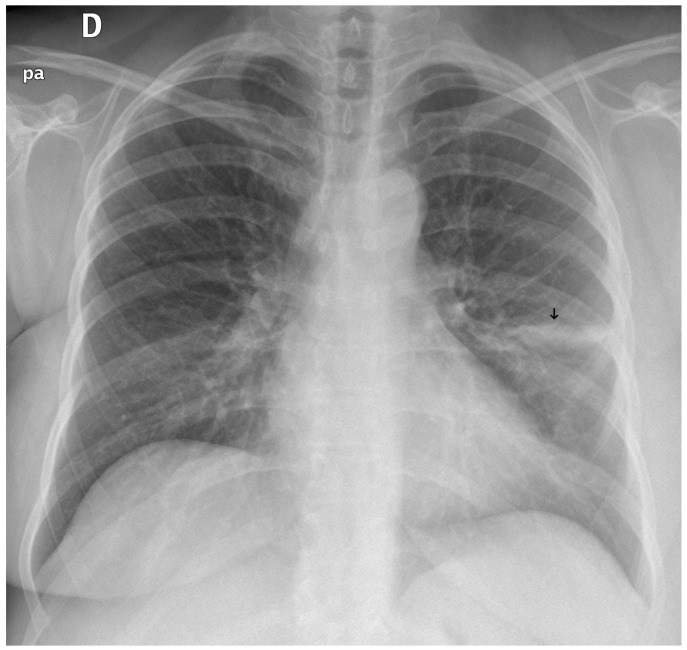
A false negative example of an alveolar opacity (black arrow).

**Figure 11 diagnostics-15-02899-f011:**
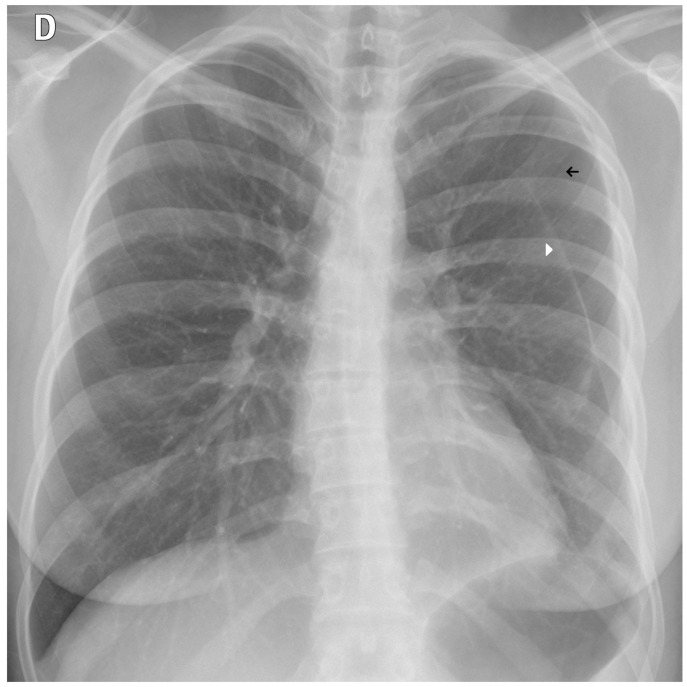
A false negative example of a missed loculated pneumothorax. The presence of pulmonary markings (black arrow) beyond the pleural line (white arrowhead) may have misled the algorithm.

**Figure 12 diagnostics-15-02899-f012:**
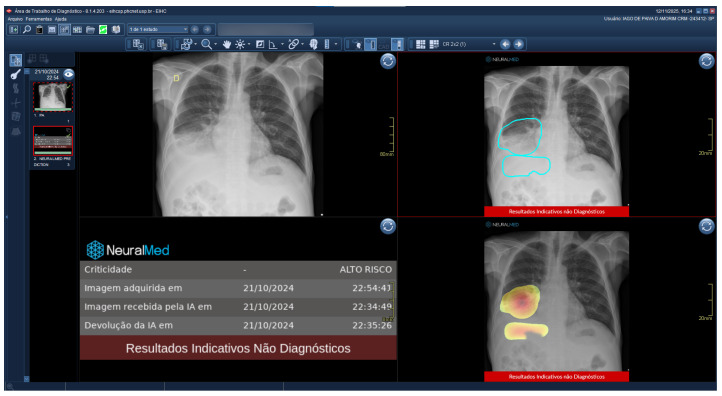
A real-world example of the TRIA system’s clinical integration within a commercial Picture Archiving and Communication System (PACS) viewer. The viewer displays the original posterior-anterior CXR (**top left**), the generated external contour image (**top right**), a heatmap visualization (**bottom right**), and the dedicated AI results panel (**bottom left**), which shows the possibility assessment and processing timestamps.

**Table 1 diagnostics-15-02899-t001:** Number of labeled cases per data source.

Source	Labeled Cases
NIH	111,783
Padchest	109,167
Private	49,866
Indiana	3783
Shenzhen	662
JSRT	246
Montgomery	138
Total Cases	275,399

**Table 2 diagnostics-15-02899-t002:** Distribution of labels in the training dataset.

Pathology	Positive Cases	Negative Cases	Total Labeled
No Finding	125,046	150,266	275,312
Opacity	32,053	240,521	272,574
Cardiomegaly	15,915	255,944	271,859
Mass/Nodule	14,527	258,383	272,910
Effusion	5392	266,535	271,927
Pneumothorax	4521	267,408	271,929
Consolidation	2483	110,920	113,403

**Table 3 diagnostics-15-02899-t003:** Performance Metrics of Individual Classifiers on the Test Set.

Pathology	Positive Cases	Sensitivity (95% CI)	Specificity (95% CI)	Accuracy (95% CI)	AUROC (95% CI)
Yesfinding	502	0.841 (0.806–0.870)	0.822 (0.789–0.851)	0.831 (0.807–0.852)	0.911 (0.893–0.927)
Cardiomegaly	199	0.960 (0.923–0.979)	0.881 (0.858–0.901)	0.895 (0.876–0.912)	0.968 (0.960–0.977)
Pneumothorax	52	0.904 (0.794–0.958)	0.879 (0.857–0.897)	0.880 (0.859–0.898)	0.955 (0.927–0.980)
Effusion	150	0.840 (0.773–0.890)	0.924 (0.905–0.939)	0.912 (0.894–0.928)	0.935 (0.907–0.959)
Opacity	264	0.848 (0.800–0.887)	0.733 (0.702–0.762)	0.761 (0.735–0.786)	0.865 (0.841–0.888)
Mass	56	0.786 (0.662–0.873)	0.862 (0.840–0.882)	0.858 (0.836–0.878)	0.839 (0.778–0.891)
Consolidation	41	0.829 (0.687–0.915)	0.656 (0.626–0.684)	0.662 (0.634–0.690)	0.775 (0.703–0.835)

Notes: AUROC = area under the receiver operating characteristic curve; CI = Confidence Interval.

**Table 4 diagnostics-15-02899-t004:** Summary of statistical comparisons among ensemble strategies. Significant results (p<0.05) are shown in bold.

Comparison	Metric	Test	*p*-Value	Significance
Threshold vs. Weighted Sum	AUROC	DeLong	**0.0307**	Yes
Threshold vs. Weighted Sum	Accuracy	McNemar	**<0.001**	Yes
Threshold vs. Weighted Sum	Sensitivity	McNemar	**<0.001**	Yes
Threshold vs. Weighted Sum	Specificity	McNemar	**<0.001**	Yes
Weighted Sum vs. Possibility	Accuracy	McNemar	**<0.001**	Yes
Weighted Sum vs. Possibility	Sensitivity	McNemar	**<0.001**	Yes
Weighted Sum vs. Possibility	Specificity	McNemar	**<0.001**	Yes
Weighted Sum vs. Hierarchical	Accuracy	McNemar	**0.0043**	Yes
Weighted Sum vs. Hierarchical	Sensitivity	McNemar	0.8450	No
Weighted Sum vs. Hierarchical	Specificity	McNemar	**<0.001**	Yes

**Table 5 diagnostics-15-02899-t005:** Performance Comparison of Four Ensemble Strategies for Abnormality Detection.

Ensemble	Accuracy (95% CI)	Sensitivity (95% CI)	Specificity (95% CI)	PPV (95% CI)	NPV (95% CI)	F1 Score	AUROC (95% CI)
Weighted Sum	0.854 (0.831–0.874)	0.845 (0.810–0.875)	0.861 (0.830–0.887)	0.836 (0.800–0.866)	0.869 (0.838–0.894)	0.840	0.927 (0.911–0.940)
Hierarchical	0.830 (0.806–0.851)	0.847 (0.812–0.876)	0.815 (0.781–0.845)	0.794 (0.756–0.827)	0.864 (0.832–0.890)	0.819	N/A
Possibility	0.793 (0.768–0.817)	0.922 (0.895–0.943)	0.685 (0.646–0.722)	0.711 (0.674–0.745)	0.913 (0.883–0.936)	0.803	N/A
Threshold	0.762 (0.735–0.787)	0.943 (0.919–0.961)	0.609 (0.568–0.648)	0.670 (0.633–0.704)	0.928 (0.897–0.950)	0.783	0.919 (0.901–0.935)

Notes: PPV = Positive Predictive Value; NPV = Negative Predictive Value; N/A = Not applicable for discrete ensemble outputs.

**Table 6 diagnostics-15-02899-t006:** Comparison of TRIA Performance with CheXNet [25].

Pathology	CheXNet AUC (95% CI)	TRIA AUROC (95% CI)
Atelectasis	0.862 (0.825–0.895)	–
Consolidation	0.893 (0.859–0.924)	0.775 (0.703–0.835)
Edema (Opacity)	0.924 (0.886–0.955)	0.865 (0.841–0.888)
Pneumonia(Opacity)	0.851 (0.781, 0.911)	0.865 (0.841–0.888)
Effusion	0.901 (0.868–0.930)	0.935 (0.907–0.959)
Cardiomegaly	0.831 (0.790–0.870)	0.968 (0.960–0.977)
Emphysema	0.704 (0.567–0.833)	–
Hernia	0.851 (0.785–0.909)	–

Notes: As pathology mappings of CheXNet and TRIA were different, Edema and Pneumonia in CheXNet were mapped to opacity. In the TRIA training dataset, these two former classes were included in the opacity class. TRIA did not classify atelectasis, Emphysema, and Hernia.

**Table 7 diagnostics-15-02899-t007:** Comparative Performance Metrics of some AI Algorithms for CXR Abnormality Detection.

Study/Algorithm	Type	AUROC	Sens. (%)	Spec. (%)
This Study				
TRIA (Possibility)	R	N/A	92.2	68.5
TRIA (Weighted Sum)	R	0.927	84.5	86.1
Blake et al., 2023 [16]				
Qure.ai qXR	R	N/A	99.6	67.4
Vasilev et al., 2023 [17]				
Lunit INSIGHT	R	0.940	90.0	89.0
Lunit INSIGHT	P	0.840	77.0	81.0
Catalina et al., 2024 [26]				
Oxipit ChestEye	P	N/A	47.0	92.0
Arzamasov et al., 2024 [28]				
Qure.ai qXR	R	0.921	N/A	N/A
Lunit INSIGHT	R	0.932	N/A	N/A
de Camargo et al., 2025 [23]				
LAM Algorithm	R	0.938	36.3	99.5
Qin et al., 2019 [27]				
Lunit INSIGHT	R	0.940	95.0 ^†^	80.0 ^†^
Qure.ai qXR	R	0.940	95.0 ^†^	72.0 ^†^

Notes: N/A = Not Available or not reported as a primary endpoint; R = Retrospective; P = Prospective; LAM = Lung Abnormality Model. ^†^ Metrics from Qin et al. represent the operating point required to achieve a sensitivity of at least 95%.

## Data Availability

The processed datasets generated and analyzed during the current study are publicly available on https://zenodo.org/records/17401230—accessed on 8 November 2025. However, the images and complete reports, which contain confidential and sensitive health data from HCFMUSP, have not been released.

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
