# Peer review of "External Validation of an Artificial Intelligence Triaging System for Chest X-Rays: A Retrospective Independent Clinical Study"

_diagnostics, 2025, doi:10.3390/diagnostics15222899_

Round 1
Reviewer 1 Report
Comments and Suggestions for Authors
In this study, it is stated that a number of tasks were performed, such as segmentation and classification of lung X-ray images, Grad-CAM analysis, and others. However, the manuscript is not yet mature enough. In particular, methodological and experimental details are insufficient. It is unclear why the U-Net architecture was chosen and how the outputs were evaluated. Additionally, the relationship between the classification architecture and Grad-CAM analyses is not clearly demonstrated. Although the purpose of using Grad-CAM analyses is stated, the current form does not adequately serve this purpose. Therefore, it appears that the article requires a comprehensive revision.
Suggestions for revisions
1)The language and explanations used in the manuscript should be reviewed. The study’s objective is not clearly stated, and the objective–outcome relationship appears weak.
2)References containing question marks (?) should be checked, and any missing citation information should be completed.
3)The relationship between the employed methods and the problem addressed by the study is not clear. Why was the U-Net model chosen? What is its advantage in this study, and in what way does it contribute to the methodology? Why is Grad-CAM used, and why are the results of its analyses not sufficiently visible in the manuscript?
4)Is the use of the LLM model aimed solely at extracting labels? The usage and contribution level of the model should be described in more detail.
5)The detailed explanation of the workflow illustrated in Figure 4 is needed. Additionally, this schematic should be redesigned to be more organized and comprehensible.
The language and explanations used in the manuscript should be reviewed. The study’s objective is not clearly stated, and the objective–outcome relationship appears weak.
Author Response
Comment 1: The language and explanations used in the manuscript should be reviewed. The study’s objective is not clearly stated, and the objective–outcome relationship appears weak.
Response 1: Agree. We have undertaken a revision of the entire manuscript to improve the quality of the English, as well as the overall clarity and readability. We have rewritten the Abstract and the final paragraph of the Introduction to state the study's objectives more explicitly and to ensure a clear, logical flow from our methods to our results and conclusions.
- On page 1, lines 6-8: "This study aimed to externally validate TRIA, a commercial AI-powered CXR triage algorithm..."
- On page 2, lines 69-75: "The specific aims were to: evaluate the overall diagnostic accuracy..."
Comment 2: References containing question marks (?) should be checked, and any missing citation information should be completed.
Response 2: We apologize for this oversight on this latex compilation error. We have reviewed, corrected, and properly formatted all citations throughout the manuscript.
Comment 3: The relationship between the employed methods and the problem addressed by the study is not clear. Why was the U-Net model chosen? What is its advantage in this study, and in what way does it contribute to the methodology? Why is Grad-CAM used, and why are the results of its analyses not sufficiently visible in the manuscript?
Response 3: Agree. We have expanded the "Algorithm Architecture" and "Model Explainability and Visualization" sections to clarify these points.
- On page 4, lines 110-114: "We chose U-Net for its proven effectiveness in medical image segmentation tasks, where high spatial precision is required. Its encoder-decoder “U” structure enabled the capture of both contextual and spatial information..."
- On page 6, lines 142-145: "Grad-CAM served a dual purpose in this context: first, it verified that the model's predictions are based on clinically relevant regions within the segmented thoracic area; second, it contributed to interpretability..."
Comment 4: Is the use of the LLM model aimed solely at extracting labels? The usage and contribution level of the model should be described in more detail.
Response 4: We described in more detail the use of the Large Language Model (LLM). Its function was a data structuring task, beyond label extraction. We have added a detailed paragraph in the "Validation Study Design" section explaining this process.
- On page 7, lines 174-182: "The contribution of the LLM extends significantly beyond simple label extraction; it functions as a data structuring engine. This was achieved by providing the model with a detailed, multi-component prompt that meticulously guided its output..."
Comment 5: The detailed explanation of the workflow illustrated in Figure 4 is needed. Additionally, this schematic should be redesigned to be more organized and comprehensible.
Response 5: This figure was excluded per request of the editorial office.
Reviewer 2 Report
Comments and Suggestions for Authors
This paper presents a study focused on independent and external validation in the field of CXR AI-based triage. The performance of a commercial AI system called TRIA was evaluated using real clinical data. It was observed that the system achieved high accuracy, particularly in distinguishing normal from abnormal. The study aimed to alleviate the current clinical burden and proposed a practical solution for this purpose. Evaluating the performance of AI systems on real-world problems fills a significant gap in the literature. Furthermore, this study was identified to have several shortcomings, including risk of sponsor bias, data diversity, and generalizability limits, as well as ethical and methodological transparency. Overall, the article was well-organized, its design was examined using robust statistical analysis methods, but it was determined that it needed revisions regarding independence, clinical applications, and sample quality:
- The introduction section of the article is very superficial. It should definitely be expanded, and its differences from existing studies should be emphasized. The article's contributions to the literature should be summarized in this section.
- Reference citations appear as "[? ]" throughout the text of the article. The citations should be reviewed in full. - The developed system has not been compared with previous or similar versions of triage algorithms. It would be beneficial to include comparisons of TRIA with equivalent models in the literature, such as CheXNet and CAD4TB.
- Technical details of the segmentation and classification processes have not been included.
- Explanations of the technical details of the external validation data should be included. Given that differences such as device variability or patient profiles may limit generalizability, detailing these limitations is essential.
- Although a test set designed with 1,045 images is statistically acceptable, it has not been evaluated whether the results overlap when compared to the actual clinical distributions of the normal/abnormal ratio.
- No explanation was provided regarding how significance was determined in statistical comparisons.
- No evaluation was made regarding how the four different ensemble strategies were created and how each was defined as "weighted."
A determination criterion should be provided for this.
- Example classification images of false-negative results in experiments can be presented.
- The study is expected to be conducted and tested using a real-time workflow. An explanation of this limitation should be included.
- The implementation of clinical integration and real-time prioritization processes can be simulated by giving an example scenario.
Author Response
Comment 1: The introduction section of the article is very superficial. It should definitely be expanded, and its differences from existing studies should be emphasized. The article's contributions to the literature should be summarized in this section.
Response 1: Agree. We have significantly expanded this section to better contextualize the problem, the promise of AI, and the critical need for external validation. We now conclude the introduction with a clear list of the study's specific aims, effectively summarizing its contributions.
- On page 2, lines 69-75: "The specific aims were to: evaluate the overall diagnostic accuracy of TRIA's general abnormality classifier... assess the performance of individual classifiers... compare four ensemble strategies... conduct an error analysis... and compare its performance with similar available tools."
Comment 2: Reference citations appear as "[? ]" throughout the text of the article. The citations should be reviewed in full.
Response 2: We apologize for this oversight on latex compilation error. We have reviewed, corrected, and properly formatted all citations throughout the manuscript.
Comment 3: The developed system has not been compared with previous or similar versions of triage algorithms. It would be beneficial to include comparisons of TRIA with equivalent models in the literature, such as CheXNet and CAD4TB.
Response 3: Agree. We have added a new comparative analysis within the Discussion section, a new table (6), comparison with CheXNet, along with Table 7 (comparison with other commercial algorithms). We did not compare TRIA to CAD4TB due to the absence of a tuberculosis-specific class.
- On page 17, lines 342-348: "Regarding the individual pathologies classification, we compared TRIA's performance with CheXNet, a well-established deep learning model... As shown in Table 6, TRIA achieved higher AUROC values for cardiomegaly and pleural effusion..."
- On page 18, Table 7: "Comparative Performance Metrics of some AI Algorithms for CXR Abnormality Detection"
Comment 4: Technical details of the segmentation and classification processes have not been included.
Response 4: Agree.We have added significant technical detail to the "Algorithm Architecture" section, including specific models, image sizes, and training parameters.
- On page 4, lines 109-121: "We employed a U-Net architecture with a MobileNetV2 backbone... We resized input images to 512×512 pixels... The model was trained for 40 epochs..."
- On page 5, lines 128-135: "...employed an InceptionResNetV2 convolutional neural network... input images were resized to 75×75 pixels... Training utilized the Adam optimizer (learning rate = 1 × 10⁻⁴)..."
Comment 5: Explanations of the technical details of the external validation data should be included. Given that differences such as device variability or patient profiles may limit generalizability, detailing these limitations is essential.
Response 5: Agree. We have expanded the "Validation Study Design" section to provide more details on the validation cohort and have added a paragraph in the Discussion acknowledging generalizability as a limitation.
- On page 7, lines 168-170: "The validation cohort consisted of adult patients with a mean age of 47.9 years (median, 52 years; standard deviation, 18.4; range, 19–82 years)."
- On page 19, lines 390-393: "We also recognize that this validation was conducted at a single tertiary academic hospital, which may limit the generalizability of our findings to other clinical contexts, such as primary care or community hospitals..."
Comment 6: Although a test set designed with 1,045 images is statistically acceptable, it has not been evaluated whether the results overlap when compared to the actual clinical distributions of the normal/abnormal ratio.
Response 6: This crucial point we have addressed adding a new paragraph in the Discussion to justify the use of an enriched (balanced) dataset versus a consecutively sampled one.
- On page 17, lines 323-334: "From a methodological standpoint, a consecutive prospective dataset could indeed provide higher distributional validity; however, the low frequency of abnormal findings in such samples can limit the precision and statistical power... Therefore, we opted for a balanced and enriched dataset... we still acknowledge that this enrichment does not replicate true disease prevalence..."
Comment 7: No explanation was provided regarding how significance was determined in statistical comparisons.
Response 7: Agree. We have added in the "Analysis" and Results section the use of statistical tests and the p-value on comparisons of strategies.
- On page 8, lines 231-233: "...we employed two complementary tests: the DeLong test for comparing areas under the AUROC curve and McNemar's test for paired categorical outcomes... A p-value < 0.05 was considered statistically significant."
Comment 8: No evaluation was made regarding how the four different ensemble strategies were created and how each was defined as "weighted." A determination criterion should be provided for this.
Response 8: Agree. We have added a detailed, numbered list in the "Analysis" section that explicitly defines the logic for each of the four ensemble strategies.
- On page 8, lines 204-229: "Possibility-based... Threshold-based... Weighted Sum-based... Hierarchical-based..."
- On page 8, lines 213-219: "To emphasize the importance of the global abnormality detector, we assigned the score from the general yesfinding classifier a weight of 2.0... This explicit up-weighting of the yes-finding output was the basis for its 'weighted' designation."
Comment 9: Example classification images of false-negative results in experiments can be presented.
Response 9: The Figures 9, 10, and 11 display false-negative cases.
Comment 10: The study is expected to be conducted and tested using a real-time workflow. An explanation of this limitation should be included.
Response 10: We have explicitly addressed this limitation in the Discussion and outlined it as a direction for future work.
- On page 19, lines 396-399: "Additionally, this study's retrospective design may overestimate algorithmic performance compared to real-time prospective use, as reported in prior work on diagnostic AI translation to practice."
- On page 19, lines 401-404: "Future work should include prospective, multi-center validation studies to assess the algorithm's performance in real-time clinical workflows and to measure its impact on downstream outcomes..."
Comment 11: The implementation of clinical integration and real-time prioritization processes can be simulated by giving an example scenario.
Response 11: To address this, we have created a "Clinical Workflow" section (3.4) and added Figure 12, which shows a real-world example of the system's real-world clinical integration.
- On page 15, lines 277-284: "Upon image acquisition, CXRs are automatically forwarded from the institution's PACS to the TRIA server... This seamless integration enables immediate worklist prioritization..."
- On page 16, Figure 12: "A real-world example of the TRIA system's clinical integration within a commercial Picture Archiving and Communication System (PACS) viewer."
Comment 12: The study has a risk of sponsor bias.
Response 12: Agree. We have added a paragraph in the Discussion to transparently acknowledge this potential conflict and detail the measures we implemented to ensure objectivity.
- On page 18, lines 376-383: "We acknowledge that the primary potential source of conflict in this study is sponsor bias. To minimize this risk, the study was initiated at the request of the participating institution to independently assess the performance... Importantly, the validation dataset originated from an external institution, entirely independent from the one involved in TRIA's development."
Round 2
Reviewer 1 Report
Comments and Suggestions for Authors
The authors have updated the article to include the suggested revisions as much as possible. Therefore, the updated version of the article can be considered acceptable.
Comments on the Quality of English LanguageThe authors have updated the article to include the suggested revisions as much as possible. Therefore, the updated version of the article can be considered acceptable.
Reviewer 2 Report
Comments and Suggestions for Authors
The authors found that the suggested corrections were fully addressed and that improvements were made. Furthermore, the authors provided satisfactory answers to the questions posed. I believe the paper requires no further improvement.